# Investigation of the Relationship between Vitamin D Deficiency and Vitamin D-Binding Protein Polymorphisms in Severe COVID-19 Patients

**DOI:** 10.3390/diagnostics14171941

**Published:** 2024-09-03

**Authors:** Lutfiye Karcıoğlu Batur, Mehmet Dokur, Suna Koç, Mehmet Karabay, Zeyneb Nur Akcay, Ezgi Gunger, Nezih Hekim

**Affiliations:** 1Department of Molecular Biology and Genetics, Faculty of Engineering and Natural Sciences, Biruni University, Istanbul 34015, Turkey; zakcay@biruni.edu.tr (Z.N.A.); ezgigunger@gmail.com (E.G.); nezihhekim@gmail.com (N.H.); 2Department of Emergency Medicine, Medical Faculty, Biruni University, Istanbul 34015, Turkey; mdokur@biruni.edu.tr; 3Department of Anesthesiology and Reanimation, Medical Faculty, Biruni University, Istanbul 34015, Turkey; skoc@biruni.edu.tr; 4Department of Infectious Disease, Private Medicana Hospitalis Bahçelievler, Istanbul 34180, Turkey; mkarabay@medicana.com.tr

**Keywords:** vitamin D deficiency, *rs7041*, *rs4588*, vitamin D-binding protein, gene polymorphism, severity of COVID-19

## Abstract

This study explores the association of vitamin D-binding protein (VDBP) gene polymorphisms, vitamin D levels, and the severity of COVID-19, including the need for intensive care unit (ICU) hospitalization. We analyzed a cohort of 56 consecutive age- and gender-matched adult COVID-19-positive patients and categorized them into three groups: outpatients with mild illness, inpatients with moderate disease, and ICU patients. We measured levels of free, total, and bioavailable 25-hydroxyvitamin D [25(OH)D], VDBP, and albumin. VDBP polymorphisms *rs5488* and *rs7041* were identified using real-time PCR. A significant proportion of ICU patients were vitamin D-deficient (56.25%) compared to outpatients (10%) and inpatients (5%) (*p* = 0.0003). ICU patients also had notably lower levels of VDBP (median: 222 mg/L) and total 25(OH)D (median: 18.8 ng/mL). Most patients carried heterozygous *rs7041* (60.7%) and wild-type *rs4588* (58.9%) genotypes. The distribution of *rs7041* SNP varied significantly among groups (*p* = 0.0301), while *rs4588* SNP distribution did not (*p* = 0.424). Heterozygous *rs4588* patients had significantly lower VDBP levels (*p* = 0.029) and reduced bioavailable 25(OH)D compared to those with wild-type *rs4588* (*p* = 0.020). Our findings indicate that VDBP gene polymorphisms, particularly *rs7041* and rs4588, are associated with vitamin D status and the severity of COVID-19. The lower VDBP levels and bioavailable vitamin D in ICU patients suggest that these genetic variants may influence disease severity and hospitalization needs. These results highlight the potential role of VDBP polymorphisms in COVID-19 severity, suggesting that genetic screening could be valuable in assessing the risk of severe outcomes and guiding personalized treatment strategies.

## 1. Introduction

As of 20 April 2022, more than 504.4 million confirmed COVID-19 cases and over 6.2 million related deaths worldwide had been reported to WHO [1]. If diagnosed patients exhibit mild symptoms, they can receive treatment without hospitalization [2,3]. If the symptoms are severe, hospitalization is recommended, and if this is not enough, intensive care unit (ICU) administration takes place [4]. Changes in serum concentration of 25-hydroxyvitamin D [25(OH)D] are associated with different diseases because vitamin D plays a role in various diseases, including viral infections such as SARS-CoV-2 [5,6].

The skin can produce vitamin D, a steroid hormone, from a cholesterol-like precursor (7-dehydrocholesterol) by exposure to sunlight, or vitamin D can be provided pre-formed in the diet. The body primarily metabolizes it into 25(OH)D for circulation and then transports it to the necessary parts [7]. ALP, which makes up 15% of circulating vitamin D, is the second most important protein for carrying the 25(OH)D molecule. The first is vitamin D-binding protein (VDBP), which makes up 85% of circulating vitamin D components. The liver primarily produces VDBP, a single-nucleotide polymorphism (SNP). The VDBP gene has more than 120 different types of polymorphisms. The most prevalent polymorphisms are Gc1f, Gc1s (rs7041 locus), and Gc2 (rs4588 locus). A study by Al-Daghri et al. looked at the effects of vitamin D supplements on the most common VDBP polymorphisms. They discovered that people with the major homozygous *rs7041* genotype had significantly higher levels of 25(OH)D. People who had homozygous major genotypes at *rs7041* and *rs4588* had higher levels of 25(OH) D after supplementation than people who had other genotypes [8]. They have different vitamin D-binding strengths. Numerous studies have demonstrated a correlation between these polymorphisms and the serum’s 25(OH)D level [9,10,11]. A previous study found a likely link between the *rs7041* and *rs4588* polymorphisms on the VDBP gene, as well as a higher risk of getting SARS-CoV-2 and dying in white people [12].

Various studies have reported the relationship between vitamin D status and COVID-19 in individuals [13,14,15,16,17,18]. In an ecological study using data from 46 countries, Mariani et al. found a positive correlation between vitamin D insufficiency and increased risk of SARS-CoV-2 infection and mortality [16]. Another ecological study using COVID-19 data from 12 European countries found a correlation between mean 25(OH)D values and increased COVID-19 mortality risks [17]. A further study with data from 20 European countries found a negative correlation between the number of COVID-19 cases and their 25(OH)D levels. However, the study found no significant correlation between 25(OH)D concentrations and COVID-19 mortality [18]. A recent study revealed a correlation between low serum levels of total 25(OH)D, a worse prognosis, and increased mortality rates in COVID-19 patients admitted to the ICU of a tertiary hospital in Turkey [15]. This correlation may be due to secondary microbial infections [15]. However, elucidating the underlying reasons for this association remains a challenge.

The goal of this genetic association study is to determine whether low vitamin D levels caused by these VDBP polymorphisms in adult COVID-19 patients necessitate ICU care and if a genetic link exists between these polymorphisms and the frequency of hospitalizations in COVID-19 patients. Our hypothesis posits that there is a significant association between the VDBP gene polymorphisms and the hospitalization status of COVID-19 patients. Specifically, individuals with certain genetic variations (rs7041 and rs4588) in the VDBP gene may have a higher likelihood of requiring hospitalization due to COVID-19 infection.

## 2. Materials and Methods

### 2.1. Patients, Subjects, and Setting

We prepared the article structure in accordance with the STREGA reporting guidelines [19]. This case–control genetic association study included consecutive adult patients (aged 20 to 80) who applied to Biruni University Hospital for any reason between 1 July 2021 and 1 May 2023 and received treatment. This study screened a total of 90 patients, selecting 56 age- and gender-matched patients based on the inclusion criteria. This study recruited patients based on criteria such as being of Turkish origin, having a positive PCR test for COVID-19, not taking vitamin D supplements in the last year, and not having a known disease affecting vitamin D metabolism. Figure 1 displays the participant flow diagram for this study. These patients were grouped as follows: (1) mild illness without hospitalization (outpatient); (2) hospitalized and having moderate disease without the need for ICU (inpatient); and (3) patients in need of ICU due to COVID-19 (ICU patients). We performed age- and gender-matching for the study groups to reduce potential bias in patient selection [20].

### 2.2. Data Sources and Measurement

We collected blood samples in tubes from patients who met this study’s inclusion criteria, as their doctors routinely request, for biochemical analyses, DNA isolation, and real-time PCR. We prepared the serum samples for analysis following the kit sample preparation protocol. We transferred the samples to tubes appropriate for the procedure and performed the necessary numbering.

Total 25(OH)D concentrations were measured by a chemiluminescence microparticle immunoassay method (CMIA), using an Architect 25 OH Vitamin D kit (5P02, Abbott Diagnosis, Chicago, IL, USA) and an i1000SR analyzer (Abbott Laboratories, Chicago, IL, USA) to find out how much 25(OH)D was in the blood.

The 2011 IOM report on dietary reference intakes of vitamin D established the threshold for total 25(OH)D at 20 ng/mL [20]. In accordance with the kit protocol, we measured the free vitamin D level using a free vitamin D ELISA kit (DIAsource Free vitamin D Immunoassay SA, Louvain-la-Neuve, Belgium). Bioavailable vitamin D was defined by adding the free 25(OH)D concentration and the ALB-bound vitamin D concentration. A report by Aloia et al. states that the reference range for free and bioavailable 25(OH)D levels is 2 pg/mL and 2 ng/mL, respectively [21]. We used the Human Vitamin VDBP Quantikine Immunoassay Kit (catalog number DVDBP0, R&D Systems, Minneapolis, MN, USA) to measure the amount of VDBP in plasma, adhering to the kit’s instructions and previous research [22]. Calcium, albumin, and phosphate concentrations were measured by a colorimetric test study (Roche/Hitachi Cobas C, Mannheim, Germany). PTH concentration was also measured by a colorimetric test study (Roche/Hitachi Cobas E, Mannheim, Germany).

We used the Quick-DNA^TM^ miniprep plus kit (Zymo Research Corp., Irvine, CA, USA) for DNA isolation from whole blood, following the instructions of the specified commercial kit. We were able to genotype genomic DNA separated for VDBP polymorphisms (*rs5488* and *rs7041* SNPs) by following the right steps and using the right TaqMan probes. Our base sequences for genotyping *rs4588* were TTGTTAACCAGCTTTGCCAGTTCC*[G/T]TGGGTGTGGCATCAGGCAATTTTC for *rs4588* and C_3133594_30 for rs7041. These sequences were generated by Applied Biosystems TaqMan SNP Genotyping Assays from Thermo Fisher Scientific™ (Gaithersburg, MD, USA).

### 2.3. Data Management

The sample size of this study was calculated using G*Power 3.1.9 (G * Power; universitat- Düsseldorf; Germany) program. When the a priori hypothesis was considered as “comparison of Vitamin D related variables according to 3 different groups”, the Type I error amount was taken as α = 0.05 (95%), the targeted power of the test was 1 − β = 0.80, and the minimum sample size required for statistical analysis was calculated as 57, considering that the difference between the 3 groups for total 25(OH)D level would be at a large level due to the lack of similar studies in the literature [23].

### 2.4. Statistical Analysis

We used the GraphPad Instat software version 3.1 (instat.exe). to conduct statistical analysis. We calculated descriptive statistics such as mean, standard deviation, median, minimum, and maximum values. Compatibility with the Hardy–Weinberg equilibrium model was determined using a Chi-square test. Using the Kolmogorov–Smirnov test, we assessed the normality of the variable distribution and the homogeneity of variance. We applied parametric analysis of variance (student *t* test) to compare two groups and used one-way analysis of variance (ANOVA), followed by a post hoc Tukey–Kramer multiple comparison test for multiple samples. When the parametric test assumptions failed, we used the Mann–Whitney U test for two-group comparisons, as well as the Kruskal–Wallis test (nonparametric ANOVA) and Dunn’s multiple comparison test for more than two groups. We analyzed categorical statistics using the Chi-square test if the observed count was more than 10. If the count was ≤10, Fisher’s exact test was used. Statistical significance was considered at *p* < 0.05 with a 95% confidence interval.

## 3. Results

Table 1 presents a comparison of the patients’ demographic characteristics and biochemical findings based on their hospitalization status and the severity of COVID-19. Of the 56 consecutive COVID-19 patients selected for this study, 28.6% were hospitalized in the ICU, 35.7% were outpatients, and 35.7% were inpatients hospitalized in the COVID-19 services of the clinic (Figure 1). The mean age of ICU patients was significantly higher than that of outpatients and inpatients (*p* < 0.0001). Among all patients, more than half were male (57.1%), and the most common comorbidity was hypertension (23.2%). The distribution of sex and comorbidities did not differ among the groups of hospitalized patients. 75% of outpatients, 20% of inpatients, and 43.75% of ICU patients were smokers (*p* = 0.0022). The mortality rate among all patients was 16.1%, all of whom were hospitalized in the ICU (*p* < 0.0001).

According to Table 1, ICU patients had a median total 25(OH)D value of 18.8 ng/mL [range: 12.9–77.7 ng/mL], which was considerably lower compared to outpatients and inpatients (*p* = 0.0105). The concentrations of free vitamin D and bioavailable vitamin D did not differ between hospitalization groups. Nine of the sixteen ICU patients were vitamin D deficient, while only two outpatients and one inpatient were vitamin D deficient (*p* = 0.0003). The median VDBP level of ICU patients was significantly lower than that of outpatients and inpatients (*p* = 0.0009). The average levels of calcium and albumin were much lower in ICU patients than in other groups (*p* = 0.0007 and *p* = 0.0001, respectively). However, the median iPTH level was much higher in ICU patients than in other groups (*p* = 0.0021). The phosphor concentrations did not differ among groups (Table 1).

Table 2 presents an evaluation of the distribution of VDBP gene polymorphisms based on hospitalization status. All VDBP SNP genotype compositions, whether hospitalized in the ICU or not, were consistent with the Hardy–Weinberg equilibrium model (*p* > 0.05). The distribution of genotypes of the *rs7041* SNP differed significantly among the groups of hospitalization (*p* = 0.0301), but those of the *rs4588* SNP did not (*p* = 0.424). Most of the patients had a heterozygous genotype of the *rs7041* SNP (60.7%). We found a wild-type *rs7041* SNP in 30% of inpatients and 25% of ICU patients but not in outpatients. We found a homozygous genotype (T > G) for the *rs7041* SNP in 30% of outpatients, 5% of inpatients, and 31.3% of ICU patients. A total of 70% of outpatients, 65% of inpatients, and 43.7% of ICU patients had the heterozygous genotype (T > G) for the *rs7041* SNP. Most of the patients had the wild-type *rs4588* SNP (58.9%). We found a wild-type *rs4588* SNP in 70% of inpatients, 55% of ICU patients, and 50% of outpatients. Only one of the ICU patients had a homozygous genotype (C > A) for the *rs4588* SNP, unlike the other groups. We found a heterozygous genotype (C > A) for the *rs4588* SNP in 30% of outpatients, 45% of inpatients, and 43.7% of ICU patients.

Table 3 presents a comparison of the vitamin D status findings according to VDBP gene polymorphisms. None of the vitamin D-related findings differed significantly between the genotypes for *rs7041* and *rs4588* SNPs (*p* > 0.05).

Table 4 presents a comparison of the vitamin D findings based on the *rs7041* polymorphism and ICU status. We combined the outpatients and inpatients in this analysis because there was no wild-type genotype among the outpatients. We found no differences in any of the parameters among the genotypes for *rs7041* SNP in outpatients, inpatients, or ICU patients (*p* > 0.05).

We also compared the vitamin D findings using the *rs4588* polymorphism based on ICU status, as shown in Table 5. We excluded homozygosity from the analysis among the outpatients and inpatients because no patient had this genotype. People with the wild-type genotype had a much higher mean level of VDBP than people who were heterozygous for the *rs4588* SNP (*p* = 0.029). This was true for both outpatients and inpatients. Other parameters did not differ among the genotypes for the *rs4588* SNP in the outpatients and inpatients (*p* > 0.05). We found that ICU patients with the wild-type genotype for *rs4588* SNP had significantly less bioavailable 25(OH)D than those with the other genotypes (*p* = 0.020). Other parameters did not differ among the genotypes for the *rs4588* SNP in the ICU patients (*p* > 0.05).

## 4. Discussion

In this study, we found that vitamin D deficiency resulting from low total 25(OH)D and VDBP levels, as well as low calcium and albumin levels, was more common in adult patients hospitalized in the ICU due to COVID-19. More importantly, we observed that the ratio of ICU hospitalizations due to COVID-19 varied based on the VDBP gene polymorphisms, particularly *rs7041* SNP genotypes. This variation may also be influenced by vitamin D status, as *rs4588* SNP genotypes differ in the mean bioavailable 25(OH)D concentration and VDBP level. Various isoforms could influence the serum concentration and availability of 25(OH)D [24]. VDBP appears to function as a reservoir for storing serum 25(OH)D, potentially extending its half-life and constricting the amount of free 25(OH)D available [21]. We propose that the VDBP gene polymorphisms could potentially influence this link, given that 90% of the 25(OH)D in blood closely correlates with VDBP.

There are empirical observational studies examining the potential link between vitamin D deficiency and COVID-19. These studies showed that the hospitalization rate and the duration of hospital stay increased in cases of low 25(OH)D concentrations or vitamin D deficiency [25,26]. There are also reports indicating that certain factors may increase both an individual’s susceptibility to contracting COVID-19 and the likelihood of experiencing more severe disease outcomes. One of these factors, vitamin D deficiency, increased the mortality risk and mortality rate [27,28,29]. We recorded all COVID-19-related deaths in our study among ICU patients. The mean age of our ICU patients was significantly higher than those of outpatients and inpatients. More than half of ICU patients had vitamin D deficiency, and we also found that their levels of VDBP, calcium, and albumin were lower.

This study’s most important contribution to the literature is our demonstration of the significant association between VDBP gene polymorphisms (rs7041 and rs4588) and the severity of COVID-19, as evidenced by the hospitalization status of patients. This finding underscores the potential role of genetic predisposition in determining the clinical course of COVID-19, which has not been widely explored in the literature. We suggest that variations in VDBP gene polymorphisms influence the quantity and functionality of vitamin D, thereby impacting the immune system. VDBP polymorphisms can affect the binding affinity and concentration of DBP in the bloodstream. This, in turn, influences the amount of bioavailable vitamin D, as DBP is the primary carrier of vitamin D metabolites. Variations in DBP can lead to differences in the levels of free (unbound) vitamin D, which is crucial for its biological activity. Reduced bioavailability due to certain DBP polymorphisms might impair the body’s ability to use vitamin D effectively, potentially weakening the immune response [12,13,14,15].

Vitamin D plays a significant role in modulating the immune system, including enhancing the pathogen-fighting effects of monocytes and macrophages and decreasing inflammation [30]. Polymorphisms in the VDBP gene may influence the efficiency of vitamin D delivery to immune cells [7]. For example, a reduced VDBP concentration or altered binding affinity could lead to lower levels of active vitamin D available to support immune function, making individuals more susceptible to severe disease [7]. Vitamin D is also known to regulate the production of cytokines, which are critical in the immune response and inflammation [31]. An impaired vitamin D pathway due to VDBP polymorphisms might result in an exaggerated inflammatory response, commonly observed in severe COVID-19 cases. This could contribute to the cytokine storm, a hyperinflammatory condition associated with severe COVID-19 [32]. Moreover, certain VDBP polymorphisms have been associated with altered immune responses, potentially affecting susceptibility to infections, including viral ones like SARS-CoV-2. The genetic variations might influence how the body responds to viral infections, including the ability to clear the virus or mitigate the associated inflammation [33].

Vitamin D exerts its effects on target tissues by interacting with the cytosolic/nuclear vitamin D receptor (VDR), which belongs to the steroid/thyroid hormone receptor family. Polymorphisms in the VDR gene can affect the responses of tissues to vitamin D. Infections caused by enveloped viruses can enhance both cell-mediated and humoral immunity. Given the function of VDR, it is possible to associate VDR polymorphisms with the encapsulated virus SARS-CoV-2 [34]. Different viral infection outcomes also correlate with variations in the VDR gene. Apaydin et al. investigated how the COVID-19 prognosis is related to variations in the VDR gene at the Fok I, Taq I, Bsm I, and Apa I genotypes, specifically in terms of vitamin D deficiency. Their study revealed that 25(OH)D levels did not correlate with the severity or mortality of COVID-19 [35]. However, studies have linked the Fok I Ff genotype to a worse disease, the Taq TT genotype to admission to the intensive care unit, and the ApaI aa genotype to death. The literature has reported the relationship between the VDBP polymorphism and COVID-19, revealing that carriers of VDBP1, a common VDBP phenotype, are less likely to contract SARS-CoV-2 and die from COVID-19 [36]. Researchers have identified the VDBP metabolism score, CYP24A1, as significantly associated with COVID-19 severity, potentially due to the SNP rs2282679. A genome-wide association study (GWAS) that compared hospitalised and non-hospitalised COVID-19 patients and looked at their genetics showed a strong link between VDBP rs2282679 and how severe the disease was [37]. In the present study, the genotype distribution of VDBP *rs7041* SNPs significantly differed among COVID-19 patients according to their hospitalization status. Also, the amounts of bioavailable 25(OH)D and VDBP in the blood are very different depending on the genotype for *rs4588* SNPs. These findings support the literature suggesting that VDBP has immunologic properties in viral infections and that VDBP gene polymorphisms may also contribute to the pathogenesis, severity, and outcome of COVID-19. However, it is biologically plausible that some variations in the VDR and VDBP genes can affect how the immune system works during COVID-19 [38]. The main limitations of this study are as follows:

This study’s small sample size and potential for false positivity may limit its findings. The study population is restricted to individuals of Turkish origin, which limits the generalizability of the findings to other ethnic groups. Given that genetic polymorphisms can vary significantly across populations, the findings may not be applicable to a broader, more diverse population. Genetic differences in how VDBP works between people might have had an effect on the results of our study, which looked at how polymorphisms affected COVID-19 patients’ need to stay in the hospital. There are also potential confounders, such as baseline health status, socioeconomic factors, and the presence of other underlying conditions (e.g., obesity, chronic diseases) that were not thoroughly investigated in this study. These factors could influence both vitamin D levels and COVID-19 severity, potentially confounding the observed associations.

## 5. Conclusions

The current data suggest that the VDBP polymorphisms *rs7041* and *rs4588* SNPs may change the link between vitamin D levels and the risk of COVID-19-related hospitalization in the ICU. This suggests that there is a genetic link between the VDBP gene and the hospitalization status of COVID-19 patients. The association between the polymorphisms and the need for hospitalization, particularly ICU admission, suggests that genetic factors may influence not only susceptibility to severe disease but also the body’s ability to respond to the infection. However, further research is necessary to understand the impact of these polymorphisms on the risk of contracting COVID-19 and its severity. It is also important to look into how VDBP might be involved in COVID-19 pathophysiology, taking into account genetic differences in the vitamin D metabolic pathway that could affect how treatments work. It is crucial to investigate the potential involvement of VDBP in COVID-19 pathophysiology, taking into account the genetic variations in the vitamin D metabolic pathway. Specifically, when investigating the relationship between vitamin D and COVID-19-related outcomes, it is crucial to consider the potential impact of VDBP polymorphisms on treatment efficacy. Longitudinal data tracking changes in vitamin D levels, DBP, and disease progression over time would have provided more robust insights into the dynamics of these associations.

## Figures and Tables

**Figure 1 diagnostics-14-01941-f001:**
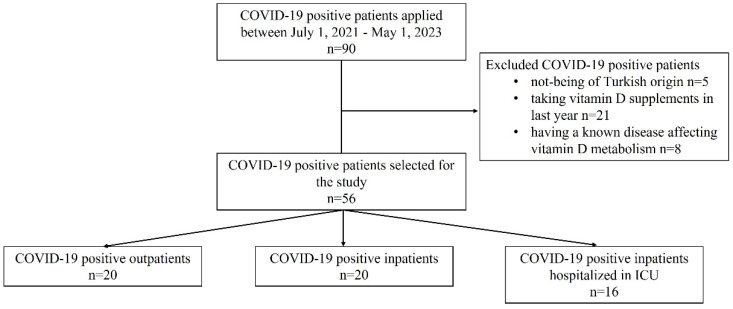
Flowchart of the study design.

**Table 1 diagnostics-14-01941-t001:** Demographical features and biochemical findings of COVID-19 patients compared by severity of COVID-19.

Variables	Total (*n* = 56)	Outpatient (*n* = 20)	Inpatient (*n* = 20)	ICU ^4^ Patient (*n* = 16)	*p*-Value
Age (years), Mean ± SD ^1^	48.87 ± 22.4	29.4 ± 9.2	48.9 ± 14.7	73.1 ± 16.4	<0.0001
Sex, *n* (%)					0.0697
Male	32 (57.1%)	13 (65%)	10 (50%)	9 (56.2%)	
Female	24 (42.9%)	7 (35%)	10 (50%)	7 (43.7%)	
Comorbidities, *n* (%)					0.149
Hypertension	13 (23.2%)	1 (5%)	5 (25%)	7 (43.7%)	
Diabetes Mellitus	10 (17.9%)	0 (0%)	4 (20%)	6 (37.5%)	
CAD ^2^	7 (12.5%)	0 (0%)	2 (10%)	5 (31.2%)	
Hypothyroidism	3 (5.4%)	2 (10%)	0 (0%)	1 (6.2%)	
Others	18 (32.1%)	4 (20%)	6 (30%)	8 (50%)	
Smoking, *n* (%)	26 (46.4%)	15 (75%)	4 (20%)	7 (43.7%)	0.0022
Mortality, *n* (%)	9 (16.1%)	0 (0%)	0 (0)	9 (56.2%)	<0.0001
Total 25(OH)D (ng/mL) Median [Min–Max]	28.9 [12.9–108.4]	28.6 [15.6–50.7]	31.4 [18.9–108.4]	18.8 [12.9–77.7] ^a^	0.0105
Free 25(OH)D (pg/mL) Median [Min–Max]	4.5 [2.04–31.17]	4.0 [2.04–6.54]	5.1 [2.74–31.17]	5.1 [3.02–14.6]	0.106
Bioavailable 25(OH)D (ng/mL) Median [Min–Max]	16.2 [6.71–101.4]	17.1 [8.77–29.27]	16.7 [8.2–101.4]	10.9 [6.71–41.35]	0.0703
Vitamin D deficient, *n* (%)	12 (21.4%)	2 (10%)	1 (5%)	9 (56.2%)	0.0003
VDBP (mg/L), Median [Min–Max]	236 [19.5–316.5]	281 [200–316]	280 [34.8–515]	222 [19.5–304] ^a,b^	0.0009
Calcium (mg/dL), Mean ± SD	8.77 ± 0.74	9.21 ± 0.42	8.70 ± 0.49 ^c^	8.32 ± 1.01 ^d^	0.0007
Albumin (g/dL), Mean ± SD	36.93 ± 9.36	45.52 ± 2.82	36.62 ± 4.61 ^d^	26.58 ± 8.44 ^d,e^	<0.0001
iPTH ^3^ (pg) Median [Min–Max]	35.1 [11.32–873.7]	30.1 [12.66–62.06]	34.0 [11.32–191]	49.0 [12.83–873.7] ^b,f^	0.0021
Phosphor (mg/dL), Median [Min–Max]	3.5 [2.18–54.53]	3.4 [2.4–4.44]	3.7 [2.44–54.53]	4.0 [2.18–6.8]	0.124

^1^ SD: Standard deviation; ^2^ CAD: coronary artery disease; ^3^ iPTH: intact parathyroid hormone; ^4^ intensive care unit; ^a^ *p* < 0.01, ^e^ *p* < 0.001; ^f^ *p* < 0.05 vs. inpatient group; ^b^ *p* < 0.01, ^c^ *p* < 0.05 and ^d^ *p* < 0.001 vs. outpatient group.

**Table 2 diagnostics-14-01941-t002:** The distribution of VDBP gene polymorphisms compared by severity of COVID-19.

Variables	Total (*n* = 56)	Outpatient (*n* = 20)	Inpatient (*n* = 20)	ICU ^1^ Patient (*n* = 16)	*p*-Value
*rs7041* SNP ^2^					0.0301
Wild type	10 (17.9%)	0 (%)	6 (30%)	4 (25%)	
Homozygote (T > G)	12 (21.4%)	6 (30%)	1 (5%)	5 (31.3%)	
Heterozygote (T > G)	34 (60.%)	14 (70%)	13 (65%)	7 (43.7%)	
*rs4588* SNP					0.424
Wild type	33 (58.9%)	14 (70%)	11 (55%)	8 (50%)	
Homozygote (C > A)	1 (1.8%)	0 (0%)	0 (0%)	1 (6.3%)	
Heterozygote (C > A)	22 (39.3%)	6 (30%)	9 (45%)	7 (43.7%)	

^1^ intensive care unit, ^2^ single-nucleotide polymorphism.

**Table 3 diagnostics-14-01941-t003:** Comparison of the findings related to vitamin D status according to VDBP gene polymorphisms.

	*rs7041*				*rs4588*		
Variables	WT (*n* = 10)	Homozygous (*n* = 12)	Heterozygous (*n* = 34)	*p*-Value	WT (*n* = 33)	Homozygous + Heterozygous (*n* = 23)	*p*-Value
Total 25(OH)D (ng/mL), Median [Min–Max]	26.9 [18.9–77.7]	28.1 [14.6–50.7]	29.5 [12.9–108.4]	0.534	28.9 [14.6–108.4]	29 [12.9–77.7]	0.696
Free 25(OH)D (pg/mL), Median [Min–Max]	4.43 [2.74–14.6]	4.28 [3.18–7.4]	4.85 [2.04–31.2]	0.784	4.34 [2.04–31.17]	4.53 [2.43–14.6]	0.973
Bioavailable 25(OH)D (ng/mL), Median [Min–Max]	15.82 [8.2–41.4]	16.12 [6.7–29.3]	16.64 [7.9–101.4]	0.659	16.45 [6.71–101.4]	16.05 [7.89–41.35]	0.474
Vitamin D deficient N (%)	1 (10%)	5 (41.7%)	6 (17.6%)	0.136	7 (21.2%)	5 (21.7%)	0.962
VDBP (mg/L), Median [Min–Max]	269 [42–515]	260.5 [19.5–307]	250 [34.8–420]	0.570	283 [19.5–420]	237.5 [80–515]	0.150
Calcium (mg/dL), Mean ± SD	8.55 ± 0.98	8.79 ± 0.95	8.83 ± 0.58	0.590	8.82 ± 0.84	8.71 ± 0.58	0.572
Albumin (g/dL), Mean ± SD	33.4 ± 8.79	36.03 ± 11.52	38.29 ± 8.64	0.291	37.84 ± 9.62	35.62 ± 9.01	0.383
iPTH ^1^ (pg) Median [Min–Max]	30.35 [12.83–75.66]	41.84 [20.08–873.7]	35.09 [11.32–191]	0.308	36.06 [11.3–873.7]	34.75 [12.8–179.6]	0.527
Phosphor (mg/dL), Median [Min–Max]	4.1 [3.07–4.71]	3.76 [2.49–5.33]	3.48 [2.18–54.53]	0.188	3.81 [2.51–3.81]	3.52 [2.18–6.8]	0.479

^1^ iPTH: intact parathyroid hormone.

**Table 4 diagnostics-14-01941-t004:** Comparison of the findings related to vitamin D status according to the *rs7041* polymorphism and ICU status.

Variables	Outpatients and Inpatients				ICU ^2^ Patients			
Parameters	WT (*n* = 6)	Homozygous (*n* = 7)	Heterozygous (*n* = 27)	*p*-Value	WT (*n* = 4)	Homozygous (*n* = 5)	Heterozygous (*n* = 7)	*p*-Value
Total 25(OH)D (ng/mL) Median [Min–Max]	26.9 [18.9–36.6]	32.9 [27.3–50.7]	29.5 [15.6–108.4]	0.268	29.6 [21.8–77.7]	17.4 [14.6–17.7]	19.9 [12.9–53.9]	0.072
Free 25(OH)D (pg/mL) Median [Min–Max]	3.94 [2.74–4.58]	4.34 [3.77–6.54]	4.53 [2.04–31.17]	0.282	10.31 [3.51–14.60]	3.47 [3.18–7.40]	5.40 [3.02–9.41]	0.170
Bioavailable 25(OH)D (ng/mL) Median [Min–Max]	14.74 [8.2–17.55]	18.38 [15.12–29.27]	16.85 [8.77–101.4]	0.078	17.08 [11.24–41.35]	8.33 [6.71–16.88]	10.56 [7.89–30.86]	0.113
Vitamin D deficient *n* (%)	1 (16.7)	0 (0)	2 (7.4)	0.523	1 (25)	0 (0)	2 (28.6)	0.428
VDBP (mg/L) Median [Min–Max]	294 [248–515]	289 [251–307]	275 [34.8–420]	0.113	232 [42–304]	200 [19.5–248]	221 [80–278]	0.794
Calcium (mg/dL) Mean ± SD	8.93 ± 0.64	9.06 ± 0.32	8.93 ± 0.55	0.845	7.99 ± 1.2	8.42 ± 1.42	8.44 ± 0.59	0.445
Albumin (g/dL) Mean ± SD	37.92 ± 4.28	44.89 ± 3.31	40.78 ± 6.28	0.091	26.63 ± 9.95	23.62 ± 4.32	28.67 ± 10.19	0.383
iPTH ^1^ (pg) Median [Min–Max]	27.72 [17.07–44.22]	29.30 [20.07–45.59]	34.69 [11.32–191]	0.413	38.96 [12.83–75.66]	146.79 [38.50–873.7]	48.98 [26.88–179.6]	0.128
Phosphor (mg/dL) Median [Min–Max]	3.95 [3.07–4.39]	3.71 [2.74–4.44]	3.44 [2.40–54.53]	0.365	4.17 [4.05–4.71]	3.95 [2.49–5.33]	3.73 [2.18–6.80]	0.683

^1^ iPTH: intact parathyroid hormone; ^2^ intensive care unit.

**Table 5 diagnostics-14-01941-t005:** Comparison of the findings related to vitamin D status according to the *rs4588* polymorphism and ICU status.

	Outpatients and Inpatients			ICU ^2^ Patients			
Variables	WT (*n* = 25)	Heterozygous (*n* = 15)	*p*-Value	WT (*n* = 8)	Homozygous *(n =* 1)	Heterozygous (*n* = 7)	*p*-Value
Total 25(OH)D (ng/mL) Median [Min–Max]	30.6 [15.6–108.4]	29.0 [16.4–42.2]	0.214	17.45 [14.6–53.9]	23.30	31.20 [12.9–77.7]	0.511
Free 25(OH)D (pg/mL) Median [Min–Max]	4.34 [2.04–31.17]	4.26 [2.43–6.98]	0.576	4.21 [3.18–14.03]	3.51	5.69 [3.02–14.60]	0.802
Bioavailable 25(OH)D (ng/mL) Median [Min–Max]	16.85 [8.77–101.4]	16.57 [8.2–24.85]	0.241	8.94 [6.71–30.86]	40.8	41.41 [12.83–179.6]	0.020
Vitamin D deficient *n* (%)	1 (4%)	2 (13.3%)	0.642	5 (62.5%)	0 (0%)	3 (42.9%)	0.440
VDBP (mg/L) Median [Min–Max]	289 [34.8–420]	247 [210.5–515]	0.029	181.5 [19.5–248]	304	223.75 [80–237.5]	0.262
Calcium (mg/dL) Mean ± SD	9.01 ± 0.56	8.86 ± 0.45	0.381	8.22 ± 1.28	7.9	8.56 ± 0.77	0.613
Albumin (g/dL) Mean ± SD	42.16 ± 5.26	39.25 ± 6.57	0.157	24.34 ± 7.29	34.3	27.78 ± 10.76	0.476
iPTH ^1^ (pg) Median [Min–Max]	33.39 [11.32–191]	31.64 [14.72–47.93]	0.696	80.21 [26.88–873.7]	40.08	39.62 [12.83–179.6]	0.312
Phosphor (mg/dL) Median [Min–Max]	3.59 [2.51–54.53]	3.32 [2.40–4.83]	0.118	4.08 [2.49–5.33]	4.05	4.21 [2.18–6.80]	0.915

^1^ iPTH: intact parathyroid hormone; ^2^ intensive care unit.

## Data Availability

The datasets generated and/or analyzed during the current study are available from the corresponding author on reasonable request.

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
