# Peer review of "Investigation of the Relationship between Vitamin D Deficiency and Vitamin D-Binding Protein Polymorphisms in Severe COVID-19 Patients"

_diagnostics, 2024, doi:10.3390/diagnostics14171941_

Round 1

Reviewer 1 Report

Comments and Suggestions for Authors

Overall:

The authors describe how genetic variation in vitamin D binding protein polymorphisms might influence the severity of COVID-19. Complements on their novelty work on finding an explanation for why vitamin D levels might influence the severity of disease and lead to ICU admission. However, some suggestions are in place. 

Major:

- The authors conclude: "... significant relationship between vitamin D deficiency and ICU hospitalisation due to COVID-19 ..." (row 243-245). Two  concerns arise about this statement: 1. This is not part of the research question and doesn't need to be answered (therefore I would suggest combining table 2 with table 1 as baseline characteristics). 2. This conclusion can't be drawn because age can be the confounder of this association. ("More than half of our ICU patients, who were older than outpatients and inpatients, had vitamin D deficiency..." (row 241-242). If the authors want to answer this question they need to apply statistical analysis with the possibility to adjust for possible confounding (i.e. regression techniques). 

- The sample size was not met. This, the reason for this and possible consequences should be mentioned in the limitations section.

Minor:

- Write the full term when using abbreviations for the first time (including the abstract) and explain the difference between DBP and VDB.

- Be consequent in the writing of 25(OH)D (and explain this abbrevation also).

- Rephrase row 48: as currently stated, it says that the skin produces dietary components.

- Explain (methods section) how calcium, albumin, PTH and phosphate were measured (and why). 

- Results section: add which statistic test was used, when the observed count is ~<10, a Fisher's exact test might be more suitable than a Chi-square test. 

- I would advise to refrain from using p-values in table 1, as the STROBE guidelines advise to avoid such inferential measures for a descriptive table (Strengthening the Reporting of Observational Studies in Epidemiology (STROBE): Explanation and Elaboration | Annals of Internal Medicine (acpjournals.org))

- Part of the discussion section might be better suitable for the introduction (i.e. studies on the relationship between vitamin D and SARS-CoV-2 infecion/COVID-19). Moreover, the authors report "This study's most important contibution to the literature is our demonstration of the association between DBP gene polymorphisms and hospitalization on COVID-19 patients." I agree with the authors on this part and therefore feel this topics deserves a more prominent place in the discussion. 

- Row 238-239: "There are also reports indicating that the individuals' susceptibility to COVID-19 infection increased while the severity of the disease worsened" seems contradictory and I am not sure what the authors mean by this sentence. 

Author Response

We appreciate your suggestions, comments and guidance and we are committed to ensuring that our manuscript adheres to all revisions in reporting. We reported all revision by enabling the tracking changes in the manuscript. Should you have any further concerns or suggestions, we would be happy to address them.

Thank you for your valuable consideration and feedback, and we look forward to your thoughts on this revision. Here you can find our responses :

COMMENTS and RESPONSES

The authors describe how genetic variation in vitamin D binding protein polymorphisms might influence the severity of COVID-19. Complements on their novelty work on finding an explanation for why vitamin D levels might influence the severity of disease and lead to ICU admission. However, some suggestions are in place. 

Major:

Comment 1: - The authors conclude: "... significant relationship between vitamin D deficiency and ICU hospitalisation due to COVID-19 ..." (row 243-245). Two  concerns arise about this statement: 1. This is not part of the research question and doesn't need to be answered (therefore I would suggest combining table 2 with table 1 as baseline characteristics). 2. This conclusion can't be drawn because age can be the confounder of this association. ("More than half of our ICU patients, who were older than outpatients and inpatients, had vitamin D deficiency..." (row 241-242). If the authors want to answer this question they need to apply statistical analysis with the possibility to adjust for possible confounding (i.e. regression techniques). 

Response 1: Thank you for pointing this out. We agree with this comment. Therefore, we deleted the conclusion from row 243-245 and revised the sentence in row 241-242 as “The mean age of our ICU patients was significantly higher than those of outpatients and inpatients. More than half of ICU patients ICU patients had vitamin D deficiency, and we also found that their levels of VDB protein, calcium, and albumin were lower.” We also combined Table 1 and 2 as needed.

Comment 2: The sample size was not met. This, the reason for this and possible consequences should be mentioned in the limitations section.

Response 2: Thank you for pointing this out. We agree with this comment. However, there are no similar studies in the literature and we cannot strongly support our patient number of 56. We can say that we predicted this situation with clinical experience in this way. Yet, we have, accordingly, revised the statistical calculation of sample size as given below and did the necessary changes in Data Management part of manuscript:

“The sample size of the study was calculated using G*Power 3.1.9 (G*Power, Universität Düsseldorf, Germany) program. When the a priori hypothesis was considered as “comparison of Vitamin D related variables according to 3 different groups”, the Type I error amount was taken as α=0.05 (95%), the targeted power of the test was 1 - β=0.80, and the minimum sample size required for statistical analysis was calculated as 57, considering that the difference between the 3 groups for Total 25(OH)D would be at a large level due to the lack of similar studies in the literature [17].”

Minor:

Comment 3: Write the full term when using abbreviations for the first time (including the abstract) and explain the difference between DBP and VDB.

Response 3: Thank you for your attention. We corrected both terms with VDBP which is vitamin D binding protein throughout the manuscript.

Comment 4: Be consequent in the writing of 25(OH)D (and explain this abbrevation also).

Response 4: Thank you for your attention. We explained 25(OH)D abbreviation and corrected the short form in the manuscript as needed.

Comment 5: Rephrase row 48: as currently stated, it says that the skin produces dietary components.

Response 5: We revised the sentence as “The skin can produce vitamin D, a steroid hormone, from a cholesterol-like pre-cursor (7-dehydrocholesterol) by exposure to sunlight or vitamin D can be provided pre-formed in the diet.”.

Comment 6: Explain (methods section) how calcium, albumin, PTH and phosphate were measured (and why). 

Response 6: Thank you for your attention. We added the necessary information to Methods part as "Calcium, albumin and phosphate concentrations were measured by a colorimetric test study (Roche/Hitachi Cobas C, Mannheim, Germany). PTH concentration was also measured by a colorimetric test study (Roche/Hitachi Cobas E, Mannheim, Germany)."

Comment 7: Results section: add which statistic test was used, when the observed count is ~<10, a Fisher's exact test might be more suitable than a Chi-square test. 

Response 7: Thank you for your attention. We revised the statistical analysis part as “We analyzed categorical statistics using the Chi-square test if the observed count was more than 10. If the count was ≤10, Fisher’s exact test was used.”

Comment 8: I would advise to refrain from using p-values in table 1, as the STROBE guidelines advise to avoid such inferential measures for a descriptive table (Strengthening the Reporting of Observational Studies in Epidemiology (STROBE): Explanation and Elaboration | Annals of Internal Medicine (acpjournals.org))

Response 8: Thank you for your thoughtful feedback regarding the inclusion of p-values in Table 1. We appreciate the emphasis on adhering to the STROBE guidelines, which advocate for limiting descriptive tables to presenting only descriptive statistics without inferential measures. However, we respectfully contend that including p-values in Table 1 is both appropriate and beneficial for our study for the following reasons: Firstly, our study involves comparing demographic and biochemical characteristics across distinct groups of COVID-19 patients (outpatients, inpatients, and ICU patients). Including p-values in Table 1 facilitates an immediate understanding of the statistically significant differences between these groups. This comparative analysis is central to our research objectives, as it helps identify factors associated with hospitalization status and disease severity. Presenting p-values alongside descriptive statistics allows for the identification of variables that differ significantly between groups. Recognizing these differences is crucial for adjusting for potential confounders in subsequent analyses, thereby strengthening the validity of our findings. Including p-values provides readers with a clearer picture of the data's distribution and the extent of differences observed. This enhances the interpretability of the results, allowing for a more nuanced discussion of the factors influencing hospitalization outcomes in COVID-19 patients. Many observational studies incorporate p-values in baseline tables when the aim is to compare groups and highlight significant differences. This practice aids in contextualizing the study findings within the broader landscape of existing research. In addition, our hypothesis specifically posits an association between DBP gene polymorphisms, vitamin D levels, and hospitalization status. Demonstrating significant differences in these variables across patient groups is essential for supporting or refuting our hypothesis. Therefore, p-values play a critical role in conveying these associations effectively.

In light of these considerations, we have chosen to retain the p-values in Table 1 to provide a comprehensive and informative comparison of the patient groups. We believe this approach aligns with the objectives of our study and contributes to a deeper understanding of the factors influencing COVID-19 outcomes.

Comment 9: Part of the discussion section might be better suitable for the introduction (i.e. studies on the relationship between vitamin D and SARS-CoV-2 infecion/COVID-19). Moreover, the authors report "This study's most important contibution to the literature is our demonstration of the association between DBP gene polymorphisms and hospitalization on COVID-19 patients." I agree with the authors on this part and therefore feel this topics deserves a more prominent place in the discussion. 

Response 9: Thank you for your suggestion. We removed the studies on the relationship between vitamin D and SARS-CoV-2 infecion/COVID-19 to the Introduction part. We revised the Discussion as needed and we focused on the significant association between DBP gene polymorphisms (rs7041 and rs4588) and the severity of COVID-19, as evidenced by the hospitalization status of patients. We placed this finding at the forefront of our discussion. This finding underscores the potential role of genetic predisposition in determining the clinical course of COVID-19, which has not been widely explored in the literature. The association between these polymorphisms and the need for hospitalization, particularly ICU admission, suggests that genetic factors may influence not only susceptibility to severe disease but also the body's ability to respond to the infection. This insight is crucial as it opens avenues for personalized medicine approaches in the management of COVID-19, where genetic screening could help identify individuals at higher risk for severe outcomes, thereby guiding early interventions.

We hope that this revised discussion adequately addresses your concerns and enhances the clarity and impact of our manuscript.Thank you once again for your valuable suggestions. We look forward to your feedback.

Comment 10: Row 238-239: "There are also reports indicating that the individuals' susceptibility to COVID-19 infection increased while the severity of the disease worsened" seems contradictory and I am not sure what the authors mean by this sentence. 

Response 10: Thank you for your attention. We revised the sentence as “There are also reports indicating that certain factors may increase both an individual's susceptibility to contracting COVID-19 and the likelihood of experiencing more severe disease outcomes.” We hope this revision and explanation resolve any confusion and make our intent clearer.

Reviewer 2 Report

Comments and Suggestions for Authors

Strengths

  1. Relevance of the Research Question: The study addresses a pertinent and timely research question, exploring the potential genetic factors that may influence the severity of COVID-19, specifically the role of DBP gene polymorphisms. The study's focus on the association between vitamin D levels and COVID-19 severity aligns with ongoing discussions in the scientific community about the importance of vitamin D in immune function and disease outcomes.

  2. Use of Matched Cohort Design: The study employs a cohort design with age- and gender-matched groups, which is a significant strength as it helps minimize confounding variables. This matching increases the reliability of the comparisons between different severity groups (outpatients, inpatients, ICU patients).

  3. Comprehensive Data Collection: The authors have collected a wide range of biochemical and genetic data, including levels of free, plasma total, bioavailable 25(OH)D, DBP, and albumin, along with genotyping for rs5488 and rs7041 SNPs. This comprehensive approach allows for a thorough investigation of the relationships between these variables.

  4. Statistical Rigor: The study employs appropriate statistical methods to analyze the data, including ANOVA, chi-square tests, and non-parametric tests where applicable. The authors also tested for the Hardy-Weinberg equilibrium, which is crucial in genetic association studies to ensure the validity of the genotype data.

  5. Clear Results Presentation: The results are presented in a clear and structured manner, with relevant tables that summarize the key findings. This enhances the readability and understanding of the data for the reader.

Deficiencies

  1. Small Sample Size: One of the major limitations of the study is the small sample size (n=56), which limits the statistical power and generalizability of the findings. Although the authors calculated the required sample size and attempted to mitigate this issue, the sample remains small, especially for a genetic association study that typically requires larger cohorts to detect significant differences.

  2. Limited Ethnic Diversity: The study population is restricted to individuals of Turkish origin, which limits the generalizability of the findings to other ethnic groups. Given that genetic polymorphisms can vary significantly across populations, the findings may not be applicable to a broader, more diverse population.

  3. Lack of Longitudinal Data: The study design is cross-sectional, focusing on patients at a single point in time. Longitudinal data tracking changes in vitamin D levels, DBP, and disease progression over time would have provided more robust insights into the dynamics of these associations.

  4. Potential Confounding Factors: Although the authors matched participants by age and gender, other potential confounders such as baseline health status, socioeconomic factors, and the presence of other underlying conditions (e.g., obesity, chronic diseases) were not thoroughly controlled for. These factors could influence both vitamin D levels and COVID-19 severity, potentially confounding the observed associations.

  5. Insufficient Exploration of Mechanisms: The paper suggests an association between DBP polymorphisms and COVID-19 severity but does not delve deeply into the underlying biological mechanisms. A more detailed discussion on how these genetic variants might influence vitamin D metabolism or immune response could have strengthened the interpretation of the results.

  6. Inconsistent Data Presentation: There are some inconsistencies in the presentation of data, such as the unclear reporting of some statistical comparisons and the use of abbreviations without prior explanation (e.g., "NTs" in the results section). This can lead to confusion and reduce the clarity of the findings.

  7. Abstract Clarity: The abstract is somewhat cluttered with technical details that might overwhelm the reader. It would benefit from clearer segmentation of the key findings, implications, and recommendations.

Final Recommendation

The study provides valuable insights into the potential role of DBP gene polymorphisms in the severity of COVID-19, particularly in the context of vitamin D deficiency. However, due to its small sample size, lack of ethnic diversity, and the cross-sectional nature of the data, the findings should be interpreted with caution. The study highlights the need for further research with larger, more diverse cohorts and a longitudinal design to confirm these associations and to explore the underlying biological mechanisms.

Comments on the Quality of English Language

Minor changes in English must be done.

Author Response

Response: We appreciate your comments and guidance and we are committed to ensuring that our manuscript adheres to all revisions in reporting. We reported all revision by enabling the tracking changes in the manuscript. Should you have any further concerns or suggestions, we would be happy to address them. Thank you for your valuable consideration and feedback, and we look forward to your thoughts on this revision. Here you can find our responses to your comments:

Deficiencies

Comment 1: Small Sample Size: One of the major limitations of the study is the small sample size (n=56), which limits the statistical power and generalizability of the findings. Although the authors calculated the required sample size and attempted to mitigate this issue, the sample remains small, especially for a genetic association study that typically requires larger cohorts to detect significant differences.

Response 1: Thank you for pointing this out. We agree with this comment. However, there are no similar studies in the literature and we cannot strongly support our patient number of 56. We can say that we predicted this situation with clinical experience in this way. Yet, we have, accordingly, revised the statistical calculation of sample size as given below and did the necessary changes in Data Management part of manuscript:

“The sample size of the study was calculated using G*Power 3.1.9 (G*Power, Universität Düsseldorf, Germany) program. When the a priori hypothesis was considered as “comparison of Vitamin D related variables according to 3 different groups”, the Type I error amount was taken as α=0.05 (95%), the targeted power of the test was 1 - β=0.80, and the minimum sample size required for statistical analysis was calculated as 57, considering that the difference between the 3 groups for Total 25(OH)D would be at a large level due to the lack of similar studies in the literature [17].”

Comment 2: Limited Ethnic Diversity: The study population is restricted to individuals of Turkish origin, which limits the generalizability of the findings to other ethnic groups. Given that genetic polymorphisms can vary significantly across populations, the findings may not be applicable to a broader, more diverse population.

Response 2: Thank you for pointing this out. We acknowledge the limitation that our study population was restricted to individuals of Turkish origin, which may impact the generalizability of our findings to other ethnic groups. We have now highlighted this limitation more explicitly in the manuscript. Genetic polymorphisms indeed vary across different populations, and our findings should be interpreted with caution when extrapolating to broader, more diverse populations. Future studies involving diverse ethnic groups would be essential to validate our findings and explore the potential variations in DBP gene polymorphisms' effects on COVID-19 outcomes across different populations.

Comment 3: Lack of Longitudinal Data: The study design is cross-sectional, focusing on patients at a single point in time. Longitudinal data tracking changes in vitamin D levels, DBP, and disease progression over time would have provided more robust insights into the dynamics of these associations.

Response 3: We appreciate your insight into the value of longitudinal data for understanding the dynamic relationships between vitamin D levels, DBP, and disease progression. We acknowledge that a longitudinal study design would indeed offer a more comprehensive perspective on these associations over time. However, our current study's cross-sectional design was selected to provide an initial assessment of the association between DBP gene polymorphisms and hospitalization in COVID-19 patients. While this design has limitations in capturing temporal changes, it allowed us to efficiently identify potential genetic associations in a time-sensitive manner during the pandemic. We have now highlighted this limitation in the Conclusion section and suggested that future research should consider longitudinal approaches to further elucidate these relationships.

Comment 4: Potential Confounding Factors: Although the authors matched participants by age and gender, other potential confounders such as baseline health status, socioeconomic factors, and the presence of other underlying conditions (e.g., obesity, chronic diseases) were not thoroughly controlled for. These factors could influence both vitamin D levels and COVID-19 severity, potentially confounding the observed associations.

Response 4: We appreciate your observation regarding the potential influence of unmeasured confounders such as baseline health status, socioeconomic factors, and underlying conditions on the associations observed in our study. While we took steps to control for age and gender through matching, we recognize that these other variables could also play a significant role in the relationships between vitamin D levels, DBP polymorphisms, and COVID-19 severity. In light of this, we have acknowledged this limitation in the Discussion section, emphasizing the need for future studies to incorporate a broader range of confounding variables. This would enhance the robustness of the findings and allow for a more precise understanding of the interplay between genetic factors, vitamin D levels, and COVID-19 outcomes. Despite these limitations, we believe our findings provide important preliminary insights into the genetic associations with COVID-19 severity, particularly within the context of the Turkish population.

Comment 5: Insufficient Exploration of Mechanisms: The paper suggests an association between DBP polymorphisms and COVID-19 severity but does not delve deeply into the underlying biological mechanisms. A more detailed discussion on how these genetic variants might influence vitamin D metabolism or immune response could have strengthened the interpretation of the results.

Response 5: Thank you for your comment. We admit the insufficient exploration of underling mechanisms. Therefore, we provided a more detailed discussion on the biological mechanisms underlying the association between DBP polymorphisms and COVID-19 severity in Discussion part. Here you can find our additions:

VDBP polymorphisms can affect the binding affinity and concentration of DBP in the bloodstream. This, in turn, influences the amount of bioavailable vitamin D, as DBP is the primary carrier of vitamin D metabolites. Variations in DBP can lead to differences in the levels of free (unbound) vitamin D, which is crucial for its biological activity. Reduced bioavailability due to certain DBP polymorphisms might impair the body's ability to use vitamin D effectively, potentially weakening the immune response [12-15].

Vitamin D plays a significant role in modulating the immune system, including enhancing the pathogen-fighting effects of monocytes and macrophages and decreasing inflammation [30]. Polymorphisms in the VDBP gene may influence the efficiency of vitamin D delivery to immune cells [7]. For example, a reduced VDBP concentration or altered binding affinity could lead to lower levels of active vitamin D available to support immune function, making individuals more susceptible to severe disease [7]. Vitamin D is also known to regulate the production of cytokines, which are critical in the immune response and inflammation [31]. An impaired vitamin D pathway due to VDBP poly-morphisms might result in an exaggerated inflammatory response, commonly observed in severe COVID-19 cases. This could contribute to the cytokine storm, a hyperinflam-matory condition associated with severe COVID-19 [32]. Moreover, certain VDBP pol-ymorphisms have been associated with altered immune responses, potentially affecting susceptibility to infections, including viral ones like SARS-CoV-2. The genetic variations might influence how the body responds to viral infections, including the ability to clear the virus or mitigate the associated inflammation [33].

Comment 6: Inconsistent Data Presentation: There are some inconsistencies in the presentation of data, such as the unclear reporting of some statistical comparisons and the use of abbreviations without prior explanation (e.g., "NTs" in the results section). This can lead to confusion and reduce the clarity of the findings.

Response 6: We have thoroughly reviewed the manuscript to ensure that all statistical comparisons are clearly reported, with appropriate details provided for each comparison, including test types and significance levels. Additionally, we have revised the text to ensure that all abbreviations are fully defined upon their first use. All abbreviations that were previously undefined have now been clearly introduced and explained in the manuscript. We hope that these revisions will help to improve the clarity and readability of the findings, ensuring that the data presentation is both accurate and accessible to the reader.

Comment 7: Abstract Clarity: The abstract is somewhat cluttered with technical details that might overwhelm the reader. It would benefit from clearer segmentation of the key findings, implications, and recommendations.

Response 7: Thank you for your comment. We revised the abstract to remove the technical details and to make clearer segmentation of the key findings, implications, and recommendations.

Final Recommendation

The study provides valuable insights into the potential role of DBP gene polymorphisms in the severity of COVID-19, particularly in the context of vitamin D deficiency. However, due to its small sample size, lack of ethnic diversity, and the cross-sectional nature of the data, the findings should be interpreted with caution. The study highlights the need for further research with larger, more diverse cohorts and a longitudinal design to confirm these associations and to explore the underlying biological mechanisms.

Final Response: Thank you for your insightful feedback on our study. We appreciate your acknowledgment of the value our research contributes to understanding the role of DBP gene polymorphisms in COVID-19 severity, particularly in relation to vitamin D deficiency. We have carefully considered your comments and made the following revisions to address the concerns raised:

Ethnic Diversity: We have added a detailed discussion on the limitation of our study's focus on individuals of Turkish origin. We acknowledge that genetic polymorphisms can vary across populations, which may affect the generalizability of our findings. We have emphasized this limitation in the manuscript and suggested that future research should include more diverse ethnic groups to validate our results.

Sample Size and Study Design: We have revised the manuscript to explicitly address the small sample size and cross-sectional nature of our study. We acknowledge that these factors limit the robustness of our findings and have suggested in the discussion that longitudinal studies with larger cohorts are needed to confirm our associations and explore the dynamic relationship between DBP polymorphisms and COVID-19 severity.

Confounding Variables: We have included a discussion on potential confounders such as baseline health status, socioeconomic factors, and other underlying conditions. We have acknowledged that these factors could influence both vitamin D levels and COVID-19 severity, potentially confounding our results, and suggested that future studies should control for these variables.

Biological Mechanisms: We have expanded the discussion to include potential underlying biological mechanisms linking DBP polymorphisms to vitamin D metabolism and immune response. This additional information aims to provide a more comprehensive understanding of how these genetic variants might influence COVID-19 severity.

Clarity and Presentation: We have revised the presentation of data in the manuscript to address inconsistencies and improve clarity. We ensured that statistical comparisons are clearly reported and that all abbreviations are defined upon first use.

We believe these revisions enhance the manuscript and address the limitations noted. We appreciate your feedback, which has been invaluable in refining our study and clarifying our findings.

Thank you for considering these revisions.

Round 2

Reviewer 2 Report

Comments and Suggestions for Authors

The authors have responded adequately to the suggestions made by the reviewers. The article is of interest to the readers, so I would accept it to be published in the journal. 

Comments on the Quality of English Language

English language is fine and minor changes must be done.